Workshop at the 6th Symposium on Advances in Approximate Bayesian Inference (non-archival), 2024 1–25

# Unity by Diversity: Improved Representation Learning in Multimodal VAEs

**Thomas M. Sutter**                                           THOMAS.SUTTER@INF.ETHZ.CH
*Department of Computer Science, ETH Zurich*

**Yang Meng**
*Department of Statistics, UC Irvine*

**Norbert Fortin**
*Department of Neuroscience, UC Irvine*

**Julia E. Vogt**
*Department of Computer Science, ETH Zurich*

**Bahbak Shahbaba**
*Department of Statistics, UC Irvine*

**Stephan Mandt**
*Department of Computer Science, UC Irvine*

## Abstract

Variational Autoencoders for multimodal data hold promise for many tasks in data analysis, such as representation learning, conditional generation, and imputation. Current architectures either share the encoder output, decoder input, or both across modalities to learn a shared representation. Such architectures impose hard constraints on the model. In this work, we show that a better latent representation can be obtained by replacing these hard constraints with a soft constraint. We propose a new mixture-of-experts prior, softly guiding each modality's latent representation towards a shared aggregate posterior. This approach results in a superior latent representation and allows each encoding to preserve information from its uncompressed original features better. In extensive experiments on multiple benchmark datasets and a challenging real-world neuroscience data set, we show improved learned latent representations and imputation of missing data modalities compared to existing methods.

## 1. Introduction

The fusion of diverse modalities and data types is paving the way for a more nuanced and comprehensive understanding of complex phenomena studied through various sources of information. Consider, for instance, the role of a medical practitioner who must combine various tests and measurements during the diagnosis and treatment of a patient. Analyzing these tests involves synthesizing shared information across different measurements and information specific to certain tests, which are equally crucial. The machine learning methods to support this decision-making process are imperative, ensuring an optimal determination for the patient's treatment. Among the existing methods, multimodal VAEs have emerged as a promising approach to jointly model and learn from weakly-supervised heterogeneous data sources.

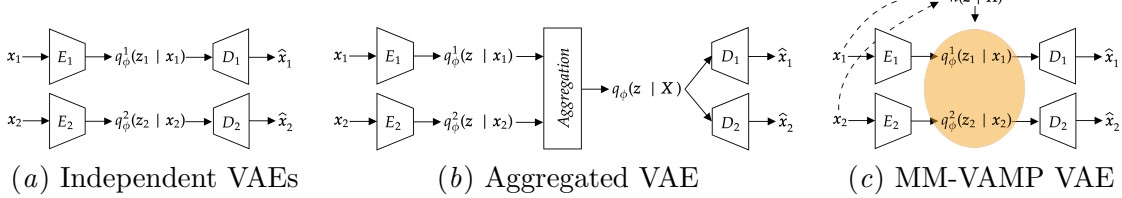

$(a)$ Independent VAEs   $(b)$ Aggregated VAE   $(c)$ MM-VAMP VAE

Figure 1: While a set of independent VAEs (fig. 1($a$)) cannot induce any multimodal capabilities, it leads to good reconstructions of the individual modalities. Multimodal VAEs with a joint posterior approximation (fig. 1($b$)) map the unimodal posterior approximations into a joint posterior approximation by aggregation. The proposed method, MM-VAMP VAE (fig. 1($c$)), builds on a set of independent VAEs but with a data-dependent prior distribution, $h(\boldsymbol{z} \mid \boldsymbol{X})$, which enables a soft-sharing of information between the different modalities.

While scalable multimodal VAEs using a single shared latent space (Wu and Goodman, 2018; Shi et al., 2019; Sutter et al., 2021) efficiently handle many modalities, finding an optimal aggregation method for the different modalities is difficult. The aggregation methods and joint representations are often sub-optimal and over-restrictive (Daunhawer et al., 2022; Sutter, 2023). This leads to inferior results regarding their learned latent representations and generative quality. There is a trade-off between the amount of shared and modality-specific information in the latent representation of the respective multimodal VAEs (Daunhawer et al., 2022; Sutter, 2023). Hence, even in simplistic scenarios, multimodal VAEs using a joint representation either suffer from limited quality or coherence in their generated samples.

In this work, we propose a novel multimodal VAE, called multimodal variational mixture-of-experts (MM-VAMP) VAE, that overcomes the limitations described above. Instead of modeling the dependency between the different modalities through a joint posterior approximation, we introduce a multimodal and data-dependent prior distribution (see fig. 1). The derivation of the proposed multimodal objective is inspired by the Vamp-prior formulation introduced by Tomczak and Welling (2017). The resulting regularization term in the VAE objective can be interpreted as a minimization between positive pairs in contrastive learning methods (Oord et al., 2019; Tian et al., 2020). We discuss these connections and links in more detail in section 3. Additionally, we show that the chosen prior distribution is optimal because it maximizes the expectation over the posterior approximation of the log-probability of the prior distribution.

## 2. Multimodal VAEs

**Problem Specification**   We consider a dataset $\mathbb{X} = \{\boldsymbol{X}^{(i)}\}_{i=1}^{n}$ where each $\boldsymbol{X}^{(i)} = \left\{\boldsymbol{x}_1^{(i)}, \ldots, \boldsymbol{x}_M^{(i)}\right\}$ is a set of $M$ modalities $\boldsymbol{x}_m$. These modalities could represent images of the same object taken from different camera angles, multiple medical measurements of a patient, text-image pairs, or—as in this paper—neuroscience data from different human or animal subjects with shared experimental conditions. When contextually clear, we remove the superscript $(i)$ in this text to remove clutter and increase readability.

We aim to learn a variational autoencoder (Kingma and Welling, 2014) for data analysis so that information among the different data modalities is shared. For example, we would like

to embed neuroscience data into a shared latent space to make brain activations comparable between different brains. At the same time, we want to avoid imposing assumptions on information sharing that are too strong to be able to take individual traits of the data modalities into account. As is typical in VAEs, this procedure involves a decoder (or likelihood) $p_\theta(\boldsymbol{X} \mid \boldsymbol{z})$, an encoder (or variational distribution) $q_\phi(\boldsymbol{z} \mid \boldsymbol{X})$, and a prior $h(\boldsymbol{z}|\boldsymbol{X})$ that we allow to depend on the input.

**Data-Dependent Prior and Objective**  The VAE framework allows us to derive a learning objective $\mathcal{E}$ as follows

$$\mathcal{E}(\boldsymbol{X}) = \mathbb{E}_{q_\phi(\boldsymbol{z}|\boldsymbol{X})} \left[ \log p_\theta(\boldsymbol{X} \mid \boldsymbol{z}) - \log \frac{q_\phi(\boldsymbol{z} \mid \boldsymbol{X})}{h(\boldsymbol{z}|\boldsymbol{X})} \right].$$

Above, $\theta$ and $\phi$ denote the learnable model variational parameters.

Importantly, our approach allows for an input-dependent prior $h(\boldsymbol{z} \mid \boldsymbol{X})$. Data-dependent priors can be justified from an empirical Bayes standpoint (Efron, 2012) and enable information sharing across data points with an intrinsic multimodal structure, as in our framework. They effectively amortize computation over many interrelated inference tasks. We stress that by making the prior data dependent, our model no longer allows for unconditional generation; however, this property can be restored by incorporating hyperpriors (Sønderby et al., 2016). We discuss the objective and its upper and lower bounds in more detail in appendix B.

**Encoder and Decoder**  We will specify our encoder and decoder assumptions. While some multimodal VAEs have a shared encoder, these approaches fail in the case of missing data modalities (Suzuki and Matsuo, 2022). Our approach assumes a separate encoder $q_\phi^m(\boldsymbol{z}_m|\boldsymbol{x}_m)$ for every modality $m$ and is more flexible. This results in a factorized posterior approximation $q_\phi(\boldsymbol{z} \mid \boldsymbol{X}) = \prod_{m=1}^M q_\phi^m(\boldsymbol{z}_m \mid \boldsymbol{x}_m)$ where $\boldsymbol{z} = [\boldsymbol{z}_1, \ldots, \boldsymbol{z}_M]$. To share information, one can then *aggregate* the different encoders into a single distribution in the latent space, e.g., by using a product or mixture of experts approach. As detailed below, this paper explores a different aggregation mechanism that preserves the individual encoders for each modality. Likewise, we assume independent decoders $p_\theta^m(\boldsymbol{x}_m|\boldsymbol{z}_m)$ for every modality $m$, assuming conditional independence of each modality given their latent representation (see also Wu and Goodman, 2018; Shi et al., 2019; Sutter et al., 2021). Note that this setup differs from contrastive learning approaches that typically involve weight sharing across modalities (Chen et al., 2020). Following this assumption, we rewrite the objective $\mathcal{E}$ as

$$\mathcal{E}(\boldsymbol{X}) = \mathbb{E}_{q_\phi(\boldsymbol{z}|\boldsymbol{X})} \left[ \log h(\boldsymbol{z} \mid \boldsymbol{X}) \right] + \sum_{m=1}^M \mathbb{E}_{q_\phi(\boldsymbol{z}_m|\boldsymbol{x}_m)} \left[ \log \frac{p_\theta^m(\boldsymbol{x}_m \mid \boldsymbol{z}_m)}{q_\phi^m(\boldsymbol{z}_m \mid \boldsymbol{x}_m)} \right]. \tag{1}$$

Our assumptions imply that the likelihood and posterior entropy terms (eq. (1), second term) decouple across modalities. In contrast, the cross-entropy between the encoder and prior (eq. (1), first term) does not decouple and may result in information sharing across modalities. We specify further design choices in the next section.

## 3. Multimodal VAMP VAE

We propose the multimodal variational mixture-of-experts (MM-VAMP) VAE, a novel multimodal VAE. Our MM-VAMP VAE draws on ideas from Tomczak and Welling (2017). The main idea is to design a mixture-of-experts prior across modalities that induces a soft-sharing of information between modality-specific latent representations rather than hard-coding this through an aggregation approach.

VAEs are an appealing model class that allows us to infer meaningful representations and generate samples with a single model. Contrastive learning approaches, on the other hand, have shown impressive results on representation learning tasks related to extracting shared information between modalities by maximizing the similarity of their representations (Radford et al., 2021). Contrastive approaches focus on the shared information between modalities, neglecting potentially useful modality-specific information. We are interested in preserving modality-specific information as it is necessary to generate missing modalities conditionally.

Therefore, we leverage the idea of maximizing the similarity of representations for generative models. We propose a prior distribution that models the dependency between the different views and a new multimodal objective that encourages similarity between the unimodal posterior approximations $q_\phi^m(\boldsymbol{z}_m \mid \boldsymbol{x}_m)$ using the regularization term in the objective as a "soft-alignment" without the need for an over-restrictive and aggregation-based joint posterior approximation. To this end, we define a data-dependent MM-VAMP prior distribution in the form of a mixture-of-experts distribution of all unimodal posterior approximations

$$h(\boldsymbol{z} \mid \boldsymbol{X}) = \prod_{m=1}^{M} h(\boldsymbol{z}_m \mid \boldsymbol{X}) \quad \text{with} \quad h(\boldsymbol{z}_m \mid \boldsymbol{X}) = \frac{1}{M} \sum_{\tilde{m}=1}^{M} q_\phi^{\tilde{m}}(\boldsymbol{z}_m \mid \boldsymbol{x}_{\tilde{m}}), \ \forall \ m \le M. \quad (2)$$

This notation implies that we use the variational distributions of all modalities $\tilde{m}$ to construct a mixture distribution and then use the same mixture distribution as a prior for any modality $m$. Finally, we build the product distribution over the $M$ components.

Our construction of a variational mixture of posteriors is similar to the VAMP-prior of Tomczak and Welling (2017) that proposes the aggregate posterior $q(\boldsymbol{z}) \equiv \frac{1}{N} \sum_{i=1}^{n} q_\phi(\boldsymbol{z} \mid \boldsymbol{x}^{(i)})$ of a unimodal VAE as a prior. Note, however, that our approach considers mixtures in *modality* space and not data space. In contrast to Tomczak and Welling (2017), our variational mixture is conditioned on $\boldsymbol{X}$ and, therefore, does not share information across data samples. Owing to this analogy, we name our new prior as "multimodal variational mixture of posteriors" (MM-VAMP) prior. Intuitively, we build the *aggregate posterior* in modality space and replicate this aggregate posterior over all modalities. We stress that this aggregate posterior differs from the standard definition as an average of variational posteriors over the empirical data distribution. Even though the prior appears factorized over the modality space, each factor still shares information across all data modalities by conditioning on the multimodal feature vector $\boldsymbol{X}$ (eq. (2)).

Figure 1 graphically illustrates the behavior of the proposed MM-VAMP-VAE compared to a set of independent VAEs and an aggregation-based multimodal VAE. A set of independent VAEs (fig. 1(a)) cannot share information among modalities. Aggregation-based VAEs

(fig. 1(b)), in contrast, enforce a shared representation between the modalities. The MM-VAMP VAE (Figure 1(c)) enables the soft-sharing of information between modalities through its input data-dependent prior $h(\boldsymbol{z} \mid \boldsymbol{X})$.

**Minimizing Jenson-Shannon Divergence**   The "rate" term $R$ in the objective, i.e., the combination of variational entropy and cross-entropy, reveals a better understanding of the effect of the mixture prior. Defining $R = KL(q_\phi(\boldsymbol{z} \mid \boldsymbol{X})||h(\boldsymbol{z}|\boldsymbol{X}))$ where $KL$ denotes the Kullback-Leibler divergence, the factorization in eq. (2) implies that

$$
R = \sum_{m=1}^{M} KL\left( q_\phi^m(\boldsymbol{z}_m \mid \boldsymbol{x}_m)||\frac{1}{M}\sum_{\tilde{m}}^{M} q_\phi^{\tilde{m}}(\boldsymbol{z}_m \mid \boldsymbol{x}_{\tilde{m}}) \right)
$$
$$
= M \cdot JS(q_\phi^1(\boldsymbol{z}_1 \mid \boldsymbol{x}_1), \ldots, q_\phi^M(\boldsymbol{z}_M \mid \boldsymbol{x}_M)), \tag{3}
$$

where $JS(\cdot)$ is the Jensen-Shannon divergence between $M$ distributions (Lin, 1991). Hence, maximizing the objective $\mathcal{E}(\boldsymbol{X})$ of the proposed MM-VAMP VAE is equal to minimizing $M$ times the JS divergence between all the unimodal posterior approximations $q_\phi^m(\boldsymbol{z}_m \mid \boldsymbol{x}_m)$. Minimizing the Jensen-Shannon divergence between the posterior approximations is directly related to pairwise similarities between posterior approximation distributions of positive pairs, similar to contrastive learning approaches but in a generative approach.

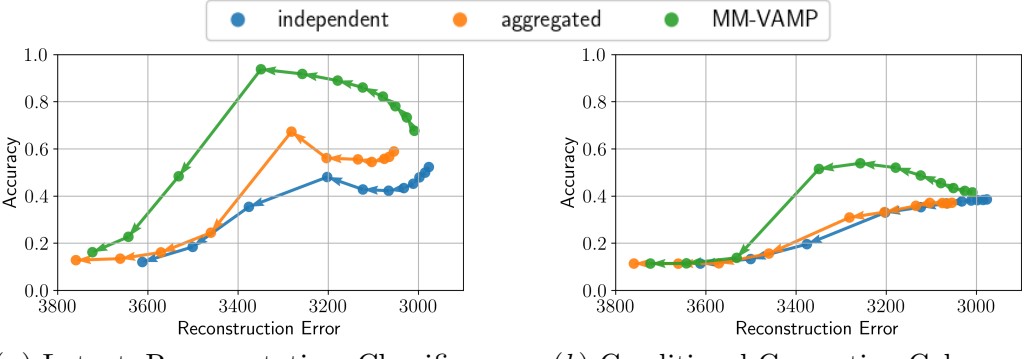

(a) Latent Representation Classification

(b) Conditional Generation Coherence

Figure 2: Results on the PolyMNIST dataset for three different VAE methods. We report the latent representation classification accuracy and the conditional generation coherence accuracy against the reconstruction error for different $\beta$ values. Every point in the figures above averages five runs over different seeds and a specific $\beta$ value where $\beta = 2^k$ for $k \in \{-8, \ldots, 3\}$. An optimal model would be in the top right corner.

### 3.1. Optimality of the MM-VAMP Prior

Theorem 1 shows that eq. (2) is *optimal* in the sense that it is the unique minimizer of the cross entropy between our chosen variational distribution and an arbitrary prior. We provide the proof in appendix B.2.

**Lemma 1**  *The expectation on the right-hand side of Equation (1) is maximized when for each $m \in \{1, \cdots, M\}$, the prior $h(\boldsymbol{z}_m|\boldsymbol{X})$ is equal to the aggregated posterior of a multimodal sample given on the first line of eq. (2).*

## 4. Experiments

We compare the proposed MM-VAMP VAE to two baseline VAE methods on three datasets. We evaluate the multimodal VAEs on the PolyMNIST (Sutter et al., 2021) in the main text. In appendices C.4 and C.5, we assess the methods on the bimodal CelebA data (Sutter et al., 2020) and present novel results on a real-world dataset measuring the hippocampal activity of a set of rats.

**Baselines** We evaluate our proposed method against two VAE formulations: a set of independent VAEs (Kingma and Welling, 2014) and a multimodal aggregated VAE (Wu and Goodman, 2018; Shi et al., 2019; Sutter et al., 2021). For the set of independent VAEs, there is no interaction or regularization between the different modalities during training. For the aggregated VAE, we use a multimodal VAE with a joint posterior approximation function. Hence, we aggregate the unimodal approximations of the different modalities and decode the modalities based on the same joint posterior approximation. We train all VAE methods as $\beta$-VAEs (Higgins et al., 2016), where $\beta$ is an additional hyperparameter weighting the rate term $R$ of the VAE (see section 3). Appendix C.3 provides the implementation details of the proposed method and the baseline alternatives. Appendix C.2 describes the evaluation procedure of our experiments.

### 4.1. PolyMNIST

**Dataset** We use PolyMNIST (Sutter et al., 2021) as a first benchmark dataset to evaluate the proposed MM-VAMP VAE against the baseline methods. The multimodal dataset uses multiple instances of the MNIST dataset (LeCun et al., 1998) with different backgrounds. Each modality consists of a digit shared between the different modalities and a modality-specific image background. It has been shown that randomly translating the digits between the modalities makes it difficult for multimodal aggregated VAEs to learn meaningful representations and coherent generation (Daunhawer et al., 2022). Figure 4 shows 10 multimodal PolyMNIST samples with random translations of the shared digit information.

**Results** Figure 2 shows the results on the translated PolyMNIST dataset. We ran the different VAE models over five seeds for every $\beta$ value and report the average performance. On this dataset, we chose $\beta \in \{2^{-8}, \ldots, 2^3\}$. Figure 2(a) shows the results of evaluating the learned latent representations against the reconstruction. The proposed MM-VAMP VAE outperforms the baseline methods, independent and aggregated VAE, by a large margin regarding the quality of the latent representation (y-axis). The reported results are the average classification accuracy of all unimodal representations. In addition, the improved latent representations do not come with a trade-off regarding the reconstruction loss (x-axis).

Figure 2(b) shows the results for evaluating the coherence of conditionally generated samples against the reconstruction loss. The proposed MM-VAMP VAE also outperforms previous works regarding the coherence of generated samples while maintaining a low reconstruction error. The reported results are the average coherence of all possible conditional generations given a single modality. Interestingly, the two curves in figs. 2(a) and 2(b) show similar dynamics for the MM-VAMP VAE but not for baseline methods, which could provide evidence for improved multimodal capabilities of the proposed method. Independent of the method, we see that there is no single $\beta$ value that leads to the best latent representation or

coherence with the lowest reconstruction error. All methods achieve the lowest reconstruction when trained with a very small $\beta$. In summary, we can show that the newly proposed MM-VAMP VAE overcomes the limitations of previous aggregation-based approaches to multimodal learning (see appendix A).

## 5. Conclusion

In this work, we have presented a new multimodal VAE, called MM-VAMP VAE, which is based on a data-dependent multimodal variational mixture-of-experts prior. By focusing on a multimodal prior, the proposed MM-VAMP VAE overcomes the limitations of previous methods with over-restrictive definitions of joint posterior approximations. We show that MM-VAMP VAE outperforms previous works on three different datasets in terms of learned latent representations as well as generative quality and coherence of missing modalities.

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

## Appendix A. Related Work

**Multimodal Learning**  While multimodal machine learning has gained attraction in recent years, mainly due to impressive results for text-to-image generation (Ramesh et al., 2021, 2022; Saharia et al., 2022), there is a long line of research on multimodal generative models (Baltrušaitis et al., 2018; Liang et al., 2022). Different from these methods, we focus on methods that are designed for a large number of modalities such that we can generate any modality from any other modality without having to train a prohibitive number of different models [1].

**Multimodal VAEs**  Multimodal VAEs that learn a joint posterior approximation of all modalities (e.g., Wu and Goodman, 2018; Shi et al., 2019; Sutter et al., 2021) require restrictive assumptions, which leads to inferior performance as shown in Daunhawer et al. (2022). Multimodal VAEs using a joint posterior approximation are based on aggregation in the latent space [2]. Extensions (Sutter et al., 2020; Daunhawer et al., 2020; Palumbo et al., 2022) have introduced modality-specific latent subspaces that lead to improved generative quality but cannot overcome limitations and require strong assumptions. A related line of work with different constraints is multiview VAEs (Bouchacourt et al., 2018; Hosoya, 2018). Unlike multimodal VAEs, multiview VAEs often use a single encoder and decoder for all views (hence, they share the parameter weights between views). While first attempts also assume knowledge about the number of shared and independent generative factors, extensions (Locatello et al., 2020; Sutter et al., 2023a,b) infer these properties during training.

**Role of Prior in VAE Formulations**  Tomczak and Welling (2017) first incorporated data-dependent priors into VAEs by introducing the VAMP-prior. Different from Tomczak and Welling (2017), who are mainly interested in better ELBO approximations, we are concerned with learning better multimodal representations and overcoming the limitations faced in previous multimodal VAE works. Sutter et al. (2020) used a data-dependent prior combined with a joint posterior approximation defining a Jensen-Shannon divergence regularization based on the geometric mean. However, their work lacks a rigorous derivation and relies on the sub-optimal conditional generation during training (see Daunhawer et al., 2022).

## Appendix B. MM-VAMP VAE

### B.1. Bound on the proposed objective

The objective $\mathcal{E}(\boldsymbol{x}_m)$ for a single modality $m$ is given by

$$\text{ELBO}_m = \mathbb{E}_{q_\phi^m(\boldsymbol{z}_m|\boldsymbol{x}_m)}\left[\log p_\theta(\boldsymbol{x}_m \mid \boldsymbol{z}_m)\right] - KL\left[q_\phi(\boldsymbol{z}_m \mid \boldsymbol{x}_m)||p_\theta(\boldsymbol{z}_m)\right] \tag{4}$$

$$\leq \mathbb{E}_{q_\phi^m(\boldsymbol{z}_m|\boldsymbol{x}_m)}\left[\log p_\theta(\boldsymbol{x}_m \mid \boldsymbol{z}_m)\right] \tag{5}$$

$$\leq \log p_\theta(\boldsymbol{x}_m \mid \boldsymbol{\mu}_m). \tag{6}$$

---

1. There are $2^M - 1$ different subsets for a dataset of $M$ modalities and, hence, paths for any-to-any mappings.

2. Sutter et al. (2021) describe how different implementations of joint posterior multimodal VAEs relate to different abstract mean definitions.

The second line (eq. (5)) follows from the non-negativity of the KL divergence. Without regularization term, the maximizing distribution is a delta distribution with zero variance (eq. (6)). Equation (6) is equal to the maximum-likelihood version of the proposed MM-VAMP VAE (for a single modality). Put differently, the MSE of a "vanilla" autoencoder is an upper bound on the ELBO for any prior distribution.

Let us have a look at when the KL actually vanishes. The KL term can only vanish if all posterior approximation $q_\phi^{\tilde{m}}(\boldsymbol{z}_m \mid \boldsymbol{x}_{\tilde{m}}$ map to a single mode $q_\phi^m(\boldsymbol{z}_m \mid \boldsymbol{x}_m$. In that case

$$h(\boldsymbol{z}_m \mid \boldsymbol{x}_m) = \frac{1}{M} \sum_{\tilde{m}=1}^{M} q_\phi^{\tilde{m}}(\boldsymbol{z}_m \mid \boldsymbol{x}_{\tilde{m}}) = q_\phi^m(\boldsymbol{z}_m \mid \boldsymbol{x}_m \tag{7}$$

and $KL\left[q_\phi^m(\boldsymbol{z}_m \mid \boldsymbol{x}_m)||h(\boldsymbol{z}_m \mid \boldsymbol{x}_m)\right] = 0$.

Another scenario is when $q_\phi^m(\boldsymbol{z}_m \mid \boldsymbol{x}_m)$ and $h(\boldsymbol{z}_m \mid \boldsymbol{x}_m)$ have disjoint modes. Hence, $q_\phi^m(\boldsymbol{z}_m \mid \boldsymbol{x}_m)$ is only a match to itself. In this case, we have

$$\mathbb{E}_{q_\phi^m(\boldsymbol{z}_m|\boldsymbol{x}_m)} \left[\log\left(\frac{1}{M} \sum_{\tilde{m}=1}^{M} q_\phi^{\tilde{m}}(\boldsymbol{z}_m \mid \boldsymbol{x}_{\tilde{m}})\right)\right] \approx \mathbb{E}_{q_\phi^m(\boldsymbol{z}_m|\boldsymbol{x}_m)} \left[\log\left(\frac{1}{M} q_\phi^m(\boldsymbol{z}_m \mid \boldsymbol{x}_m)\right)\right] \tag{8}$$

$$= -\log M + \mathbb{E}_{q_\phi^m(\boldsymbol{z}_m|\boldsymbol{x}_m)} \left[\log q_\phi^m(\boldsymbol{z}_m \mid \boldsymbol{x}_m)\right] \tag{9}$$

So our objective will be still (up to a constant) the maximum liklihood objective, leading the variances to shrink to zero. In this case, the objective will reduce to the limiting independent VAE, where the modalities do not "talk" to each other and there is no multimodal alignment.

In the most interesting case, there will be a non-trivial overlap between $q_\phi^m(\boldsymbol{z}_m \mid \boldsymbol{x}_m)$ and $h(\boldsymbol{z}_m \mid \boldsymbol{x}_m)$, i. e. between the different unimodal posterior approximations, leading to a multimodal alignment through the soft sharing that we wish to see.

In addition, fig. 3 empirically shows that the negative mean squared error (MSE) of the vanilla autoencoder (AE) upper bounds the proposed objective $\mathcal{E}$. We show that lowering the $\beta$ value of the regularizer $R$ in eq. (3) approximates the negative MSE of the vanilla AE.

## B.2. Proof of Theorem 1

**Proof** Since the cross-entropy term in eq. (1) involves an expectation over the data $\boldsymbol{X}$ and both $q_\phi(\boldsymbol{z} \mid \boldsymbol{X})$ and $h(\boldsymbol{z} \mid \boldsymbol{X})$ depend on $\boldsymbol{X}$, we can prove the identity for a given value of $\boldsymbol{X}$.

We exploit the factorization of both the variational posterior and the prior over the modalities. Interpreting the cross-entropy between the variational distribution and prior as a functional $F$ of the prior $h$, we have

$$F[h(\boldsymbol{z}|\boldsymbol{X}))] \equiv \mathbb{E}_{q_\phi(\boldsymbol{z}|\boldsymbol{X})}\left[\log h(\boldsymbol{z}|\boldsymbol{X}))\right] = \mathbb{E}_{\prod_{m=1}^{M} q_\phi^m(\boldsymbol{z}_m|\boldsymbol{x}_m)}\left[\log \prod_{m=1}^{M} h(\boldsymbol{z}_m|\boldsymbol{X}))\right]$$

$$= \sum_{m=1}^{M} \mathbb{E}_{q_\phi^m(\boldsymbol{z}_m|\boldsymbol{x}_m)}\left[\log h(\boldsymbol{z}_m|\boldsymbol{X}))\right] = M \cdot \mathbb{E}_{\frac{1}{M} \sum_{\tilde{m}=1}^{M} q_\phi^{\tilde{m}}(\boldsymbol{z}_m|\boldsymbol{x}_{\tilde{m}})}\left[\log h(\boldsymbol{z}_m|\boldsymbol{X}))\right].$$

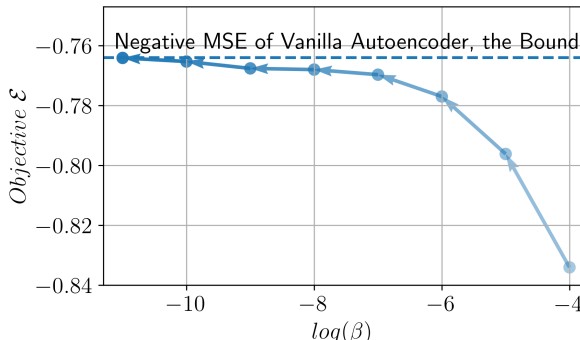

Figure 3: We compare the achieved values of the proposed objective $\mathcal{E}$ to the vanilla Autoencoder's negative mean squared error (MSE). Lowering the $\beta$ value of the regularizer $R$ in the objective in eq. (3) approximates the negative MSE bound provided by the vanilla AE. This proves empirically that the negative MSE of the vanilla AE indeed upper bounds the proposed objective $\mathcal{E}$.

As a result, we see that $F[h(\cdot)]$ is an expectation over a mixture distribution. We can solve for the optimal distribution $h(\cdot)$ by adding a Lagrange multiplier that enforces $h(\cdot)$ normalizes to one:

$$\max_{h(\boldsymbol{z}_m|\boldsymbol{X})} \underbrace{F[h(\boldsymbol{z}_m \mid \boldsymbol{X})] + \gamma \left( \int h(\boldsymbol{z}_m \mid \boldsymbol{X}) d\boldsymbol{z}_m - 1 \right)}_{\mathcal{L}[h,\gamma]}$$

To maximize the Lagrange functional, we compute its (functional) derivatives with respect to $h(\boldsymbol{z}_m|\boldsymbol{X})$ and $\gamma$.

$$\frac{\partial \mathcal{L}[h(\boldsymbol{z}_m|\boldsymbol{X}), \gamma]}{\partial h(\boldsymbol{z}_m|\boldsymbol{X})} = \frac{\frac{1}{M} \sum_{\tilde{m}=1}^{M} q_\phi^{\tilde{m}}(\boldsymbol{z}_m|\boldsymbol{X})}{h(\boldsymbol{z}_m|\boldsymbol{X})} + \gamma \overset{!}{=} 0$$
$$\frac{\partial \mathcal{L}[h(\boldsymbol{z}_m|\boldsymbol{X}), \gamma]}{\partial \gamma} = \int_{\boldsymbol{z}_m} h(\boldsymbol{z}_m|\boldsymbol{X}) d\boldsymbol{z}_m - 1 \overset{!}{=} 0$$

The first condition implies that for *any* value of $\boldsymbol{z}_m$, the ratio between the mixture distribution and the prior is constant, while the second condition demands that the prior be normalized. These conditions can only be met if the prior *equals* the mixture distribution, which proves the claim. ∎

## Appendix C. Experiments

### C.1. Implementation (General)

We use the scikit-learn (Pedregosa et al., 2011) package for the linear classifiers to evaluate the learned latent representations. All code is written using Python 3.11, PyTorch (Paszke et al., 2019) and Pytorch-Lightning (Falcon and The PyTorch Lightning team, 2019).

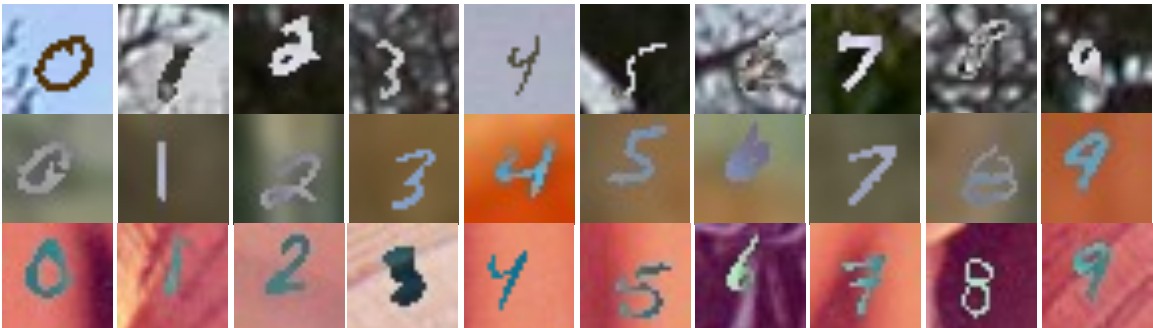

Figure 4: PolyMNIST (translated, scale=75%): every column is a multimodal tuple $\boldsymbol{X}$, and every row shows samples of a single modality $\boldsymbol{x}_m$. We see the random translation between samples by looking at images from a single row or column.

## C.2. Evaluation

We evaluate the different methods based on the coherence of their imputed samples, the quality of their latent representations, and their reconstruction error. We assume access to the full set of modalities during training, but we do not make this assumption at test time. Hence, there is a need for methods that can conditionally generate samples of these missing modalities, given the available modalities. In other words, we want to be able to impute missing modalities. Imputed modalities should not only be of high generative quality but also display the same shared information as the available modalities. For example, if we generate a sample of modality $\boldsymbol{x}_2$ based on modality $\boldsymbol{x}_1$ from the PolyMNIST dataset (see Figure 4), the generated sample of modality $\boldsymbol{x}_2$ should contain the same digit information as modality $\boldsymbol{x}_1$ but show the background of modality $\boldsymbol{x}_2$. The coherence of conditionally generated samples shows how well the content of the imputed modalities aligns with the content of the available modalities in terms of the shared information. We evaluate the coherence using ResNet-based classifiers (He et al., 2016) that are trained on samples from the original training set of every modality. In addition, we assess the learned representations based on subsets of modalities and not the full set. The quality of the representations serves as a proxy of how useful the learned representations are for additional downstream tasks that are not part of the training objective. Hence, high-quality representations of subsets of modalities are also the basis for conditionally generating coherent samples. We assess their quality by using the classification performance of linear classifiers. We train independent classifiers on the unimodal representations of the training set and evaluate them on unimodal test set representations. The reconstruction error is a proxy for how well every method learns the underlying data distribution. We evaluate the different VAE models by their achieved reconstruction error against either the learned latent representation classification or coherence (figs. 2, 6 and 11). We do this for multiple values of $\beta$, where the average performance over multiple seeds with a single $\beta$ value leads to a scatter point in the figures. This way, we can assess the trade-off between accurately reconstructing data samples and inferring shared information. In all figures, the arrows go from small to large values of $\beta$. See Appendix C.3 for more details on the evaluation metrics.

## C.3. PolyMNIST

### C.3.1. DATASET

The dataset is based on the original MNIST dataset (LeCun et al., 1998). Compared to the original dataset, the digits are scaled down by a factor of 0.75 such that there is more space for the random translation. In its original form, the PolyMNIST consists of 5 different modalities. We only use the first three modalities in this work. The background of every modality $\boldsymbol{x}_m$ consists of random patches of size $28 \times 28$ from a large image. And the digit is placed at a random position of the patch. We refer to the original publication (Sutter et al., 2021) for details on the background images. Using this setup, every modality has modality-specific information given by its background image and shared information given by the digit, which is shared between all modalities. An additional difficulty compared to the original PolyMNIST is the random translation of the digits. The dataset can be found at https://github.com/thomassutter/MoPoE.

### C.3.2. IMPLEMENTATION & TRAINING

We use the same network architectures for all multimodal VAEs. Every multimodal VAE consists of a Resnet-based encoder and a Resnet-based Decoder (He et al., 2016). All modalities share the same architecture but are initialized differently. We assume gaussian distribution for all unimodal posterior approximations, i.e.

$$q_\phi^m(\boldsymbol{z}_m \mid \boldsymbol{x}_m) = \mathcal{N}(\boldsymbol{z}_m; \boldsymbol{\mu}_m, \boldsymbol{\sigma}_m \boldsymbol{I}), \tag{10}$$

where the parameters $\boldsymbol{\mu}_m$ and $\boldsymbol{\sigma}_m$ are inferred using neural networks such that we have

$$q_\phi^m(\boldsymbol{z}_m \mid \boldsymbol{x}_m) = q_\phi^m(\boldsymbol{z}_m; \boldsymbol{\mu}_m(\boldsymbol{x}_m), \boldsymbol{\sigma}_m(\boldsymbol{x}_m)) = \mathcal{N}(\boldsymbol{z}_m; \boldsymbol{\mu}_m(\boldsymbol{x}_m), \boldsymbol{\sigma}_m(\boldsymbol{x}_m)) \tag{11}$$

The conditional data distributions $p_\theta(\boldsymbol{x}_m \mid \boldsymbol{z}_m)$ are modelled using the Laplace distribution, where the location parameter is modelled with a neural network (decoder) and the scale parameter is set to 0.75 (Shi et al., 2019), i. e.

$$p_\theta(\boldsymbol{x}_m \mid \boldsymbol{z}_m) = \mathcal{L}(\boldsymbol{x}_m; \boldsymbol{\mu}_m, \boldsymbol{\sigma}_m), \tag{12}$$

where $\mathcal{L}(\cdot)$ defines a Laplace distribution. It follows that

$$p_\theta(\boldsymbol{x}_m \mid \boldsymbol{z}_m) = p_\theta(\boldsymbol{x}_m; \boldsymbol{\mu}_m(\boldsymbol{z}_m), \boldsymbol{\sigma}_m) = \mathcal{L}(\boldsymbol{x}_m; \boldsymbol{\mu}_m(\boldsymbol{z}_m), \boldsymbol{\sigma}_m) \tag{13}$$

We use the method of Hosoya (2018) for the implementation of the aggregated VAE. In this approach, a simplistic version of joint posterior distribution is chosen where for Gaussian distribution joint posterior approximation $\mathcal{N}(\boldsymbol{\mu}_s, \boldsymbol{\sigma}_s \boldsymbol{I})$ we have the following distribution parameters $\boldsymbol{\mu}_s$ and $\boldsymbol{\sigma}_s$

$$\boldsymbol{\mu}_s = \frac{1}{M} \sum_{m=1}^{M} \boldsymbol{\mu}_m \tag{14}$$

$$\boldsymbol{\sigma}_s = \frac{1}{M} \sum_{m=1}^{M} \boldsymbol{\sigma}_s \tag{15}$$

$\boldsymbol{\mu}_m$ and $\boldsymbol{\sigma}_m$ are the distribution parameters of the unimodal posterior approximations $\mathcal{N}(\boldsymbol{\mu}_m, \boldsymbol{\sigma}_m \boldsymbol{I})$.

During training and evaluation, no weight-sharing takes place, i.e. every modality has its own encoder and decoder. We use the same architectures as in (Daunhawer et al., 2022). For all experiments on this dataset, we use an Adam optimizer (Kingma and Ba, 2014) with an initial learning rate of 0.0005, and a batch size of 256. We train all models for 500 epochs.

To assess the learned latent representation, we train a logistic regression classifier on 10000 latent representations of the training set. The accuracy is computed on all representations of the test set. To compute the coherence of conditionally generated samples, we train additional deep classifiers on original samples of the training set. We use a Resnet-based non-linear classifier that is trained on the full original training set. The prediction of this classifier is then used to determine the class (digit) of the conditionally generated sample of a missing modality. The nonlinear classifier reaches an accuracy of above 98% on the original test set. Hence, it serves as a good oracle for determining the digit of generated samples. Using the described procedure, generated samples have to be visually similar to original samples to have high coherence. Otherwise, the nonlinear classifier will not be able to predict digits correctly.

### C.3.3. Additional Results

We generated Figure 2 by plotting the classification accuracy of a linear classifier, which we trained on the learned latent representation, against the reconstruction error on the test set. For the learned latent representation, we train a classifier on the unimodal latent representations. For the aggregated VAE, this means that we train the classifier on samples of the unimodal posterior approximations and not the joint posterior approximations. Using this procedure, we test the different methods according to their performance in case of missing data, e.g., we only have access to a single modality instead of the full set at test time. However, we computed the error for the reconstruction loss given the full set of modalities. The idea for fig. 2 is to compare the reconstruction quality (how well can we learn the data distribution?) against metrics that are related to the "generative factors" of the data and relate the different modalities to each other, i. e. the shared information of a multimodal dataset.

In fig. 5, we evaluate the performance of individual modalities in case of missing modalities. For that, we reconstruct every modality if it was the only modality available at test time. Hence, the modalities in the aggregated VAE have to be reconstructed based on the unimodal posterior approximations and not the joint posterior approximation. For the independent VAEs and the MM-VAMP-VAE, the reconstruction of a modality is only based on its own unimodal posterior approximation. Hence, for the latter two methods, nothing changes in this setting. The performance of the learned latent representation and the coherence of generated samples are evaluated in the same way as in fig. 2.

Figure 5 shows that the reconstruction error of the aggregated VAE increases a lot if every modality needs to reconstruct itself. Interestingly, we can see that the "self-reconstruction error" (the x-axis in figs. 5($a$) and 5($b$)) decreases with an increasing $\beta$-value, which is different to the other two methods and also different to the aggregated VAE's behaviour in fig. 2.

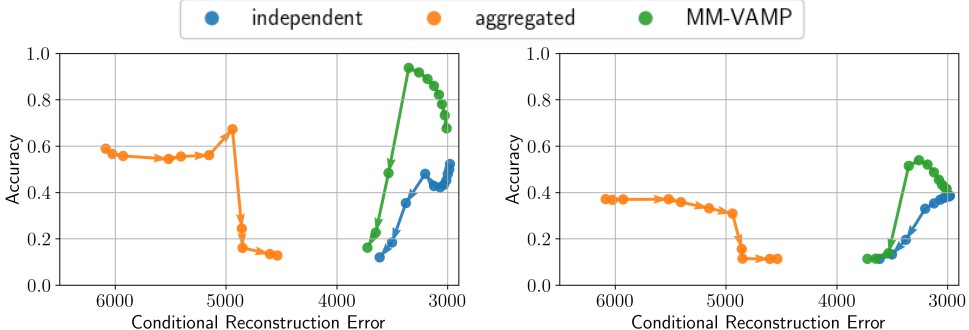

($a$) Latent Representation Classification

($b$) Conditional generation coherence

Figure 5: Results on the PolyMNIST dataset for three different VAE methods. We report the performance of the latent representation classification and the conditional generation coherence against the conditional reconstruction loss for different $\beta$ values. Every point in the figures above is the average of five runs over different seeds and a specific $\beta$ value where $\beta = 2^k$ for $k \in \{-8, \ldots, 3\}$. Different to fig. 2, the x-axis is the sum of the self-reconstruction losses if only a single modality is given as input. Hence, for the aggregated VAE, every modality is decoded by its own unimodal posterior approximation instead of the joint posterior approximation.

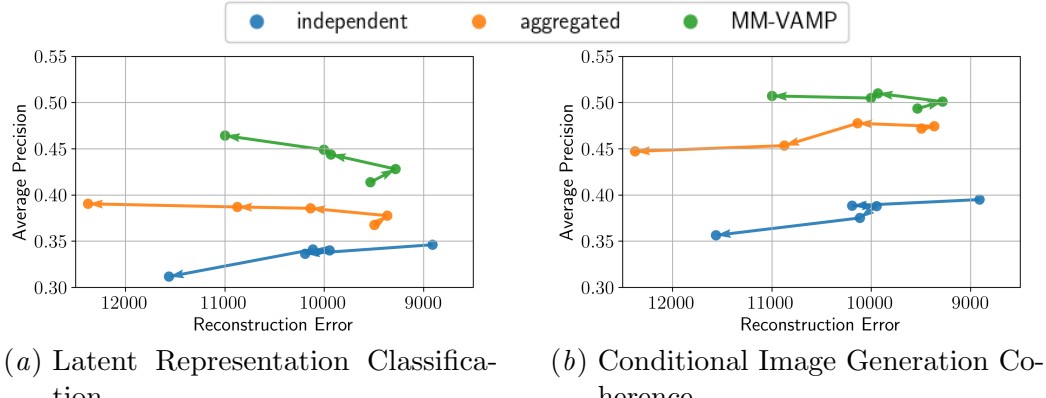

($a$) Latent Representation Classification

($b$) Conditional Image Generation Coherence

Figure 6: Results on the bimodal CelebA dataset for the three different VAE methods: independent VAEs, aggregated VAE, and the MM-VAMP VAE. We report the performance of the latent representation classification and the conditional generation coherence against the reconstruction error for different $\beta$ values. The proposed MM-VAMP VAE outperforms the baseline methods regarding the learned latent representations and the coherence of generated samples while achieving a similar reconstruction error. Every point in the figures above is the average of five runs over different seeds and a specific $\beta$ value where $\beta = 2^k$ for $k \in \{-2, \ldots, 2\}$. An optimal model would be in the top right corner.

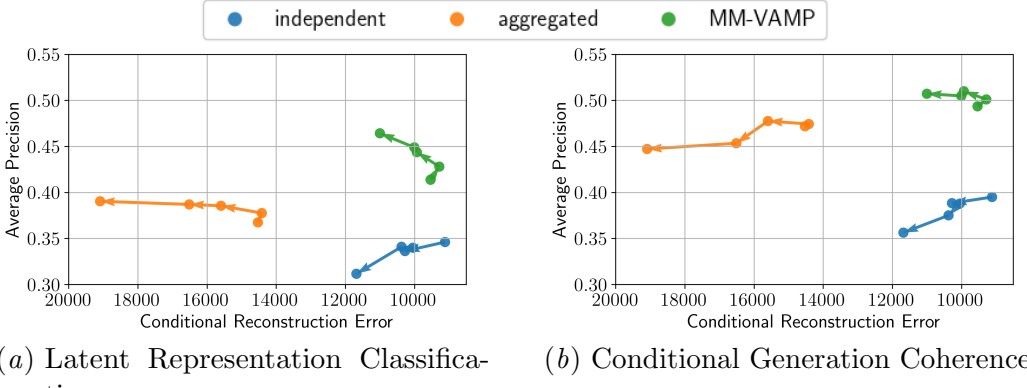

(*a*) Latent Representation Classification

(*b*) Conditional Generation Coherence

Figure 7: Results on the bimodal CelebA dataset for three different VAE methods. We report the performance of the latent representation classification and the conditional generation coherence against the conditional reconstruction loss for different $\beta$ values. Every point in the figures above is the average of five runs over different seeds and a specific $\beta$ value where $\beta = 2^k$ for $k \in \{-2, \ldots, 2\}$. Different from fig. 6, the x-axis is the sum of the self-reconstruction losses if only a single modality is given as input. Hence, for the aggregated VAE, every modality is decoded by its own unimodal posterior approximation instead of the joint posterior approximation.

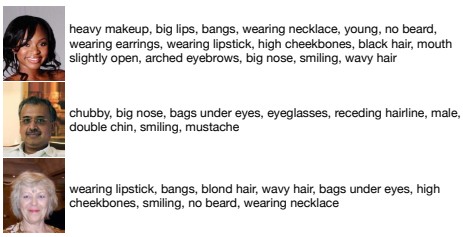

Figure 8: Bimodal CelebA: three samples of image-text pairs. To introduce another level of difficulty to the text modality we added a random translation to the starting point of the text attributes.

## C.4. Bimodal CelebA

### C.4.1. Dataset

The original CelebA data (Liu et al., 2015) is an image-based dataset of faces. We use its bimodal version with an additional text modality introduced in Sutter et al. (2020), where the text modality is generated from the 40 attributes describing every face. Compared to the PolyMNIST dataset (section 4.1), it provides more details to evaluate the different methods. It is a multi-label dataset that can compare the quality of the learned latent representations and the coherence of generated samples based on multiple attributes. In addition, some of these attributes are minor image details, which makes it more challenging to learn them within a weakly-supervised setting. A main challenge for the text modality is dealing with missing attributes, as well as the random translation of the start index for the

text string (see fig. 8). The difficulty of the individual attributes is not only given by their visual appearance but also their frequency in the dataset.

### C.4.2. IMPLEMENTATION & TRAINING

We use Resnet-based encoders and decoders for this experiments as well (He et al., 2016), similar to the ones in the PolyMNIST experiment. The image encoder and decoder consist of 2d convolutions while the text encoder and decoder consist of 1d convolutions. We use a character-level encoding and not a word or token-level encoding because of the synthetic nature of the text modality. We also use an Adam optimizer (Kingma and Ba, 2014) with a starting learning rate of 0.0002 and a batch size of 256. We train all models for 250 epochs. The implementation follows the one described in appendix C.3.

### C.4.3. RESULTS

Figure 6 shows that the good results on the PolyMNIST dataset (see fig. 2) indeed translate to the more difficult bimodal CelebA dataset. We ran the different VAE models over 3 seeds for every $\beta$ value and report the mean average precision (AP; the area under the precision-recall curve) for both the coherence and learned latent representation classification. On this dataset, we chose $\beta \in \{2^{-2}, \ldots, 2^2\}$. The proposed MM-VAMP VAE outperforms both baseline methods regarding the learned latent representation (fig. $6(a)$) while achieving a similar reconstruction error. We report the average classification performance of all unimodal latent representations. We provide additional results in appendix C.4 (fig. 9), where we evaluate the latent representations regarding the different attributes of the CelebA dataset. Our MM-VAMP VAE learns more meaningful representations with respect to almost all attributes. Figure $6(b)$ shows that the achieved coherence from conditional image generation is higher for the proposed method than previous works. For this dataset, we only conditionally generate images based on a text or image input, and the reported results are the average of the individual coherence values. Again, compared to previous works, the MM-VAMP VAE achieves higher performance metrics while reaching a similar reconstruction error. In summary, fig. 6 shows that the MM-VAMP VAE improves over previous work in terms of the learned latent representations and the coherence of generated samples. Hence, the proposed method performs well on synthetic datasets like PolyMNIST and a more complicated and difficult dataset like bimodal CelebA.

Given the multilabel nature of the CelebA dataset, we evaluate the learned latent representation also with respect to the individual attributes and not only the average performance across all attributes. Figure 9 shows the detailed results according to the full set of 40 attributes for the three methods, independent VAEs, aggregated VAE, and MM-VAMP VAE. We train again linear binary classifiers on inferred representations of the training set and evaluate them on representations of the test set. However, we now report the individual performance of every classifier. In the main text (see fig. 6), the report the average performacne of the 40 binary classifiers.

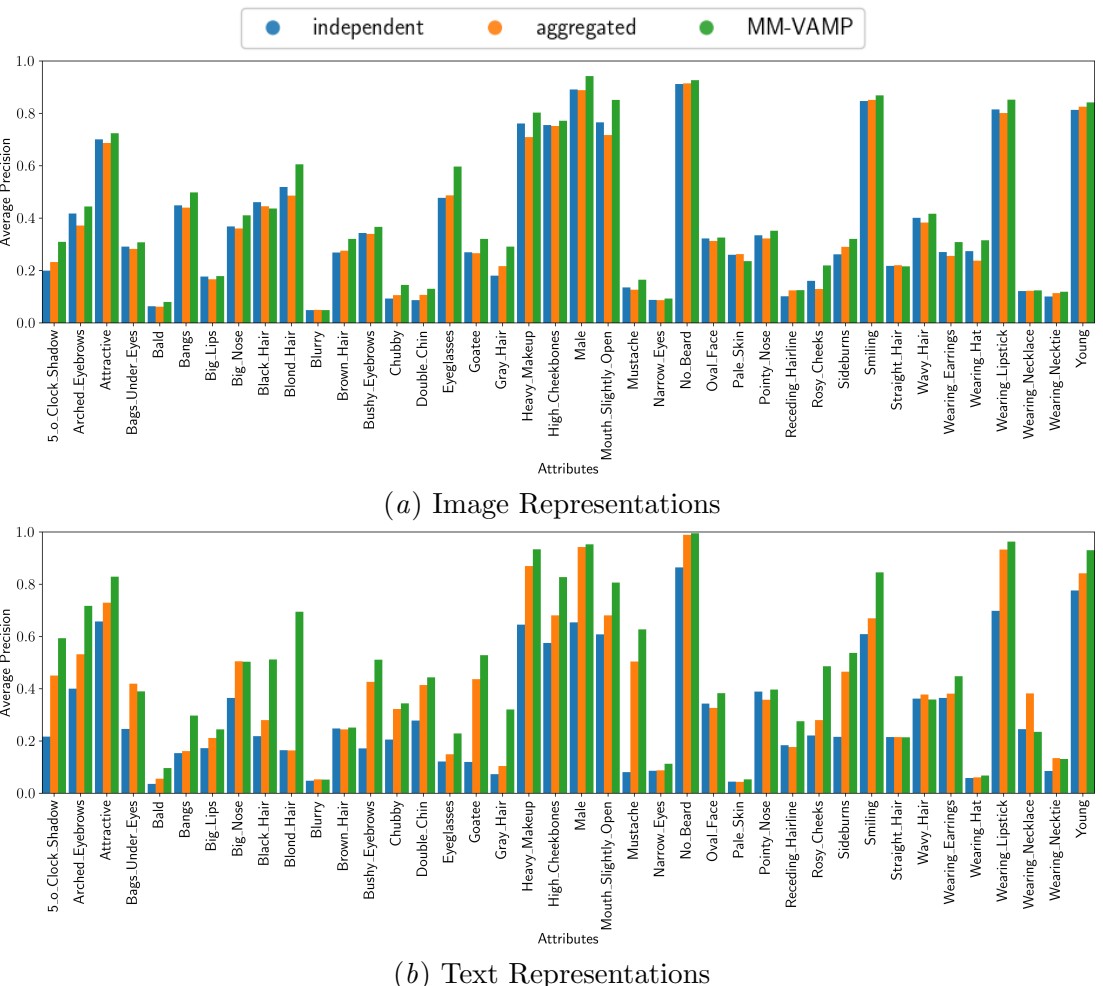

(*a*) Image Representations

(*b*) Text Representations

Figure 9: Attribute-level results on the bimodal CelebA datasets for the latent representation classification. The MM-VAMP VAE outperforms the independent VAEs and the aggregated VAE on most of the attributes.

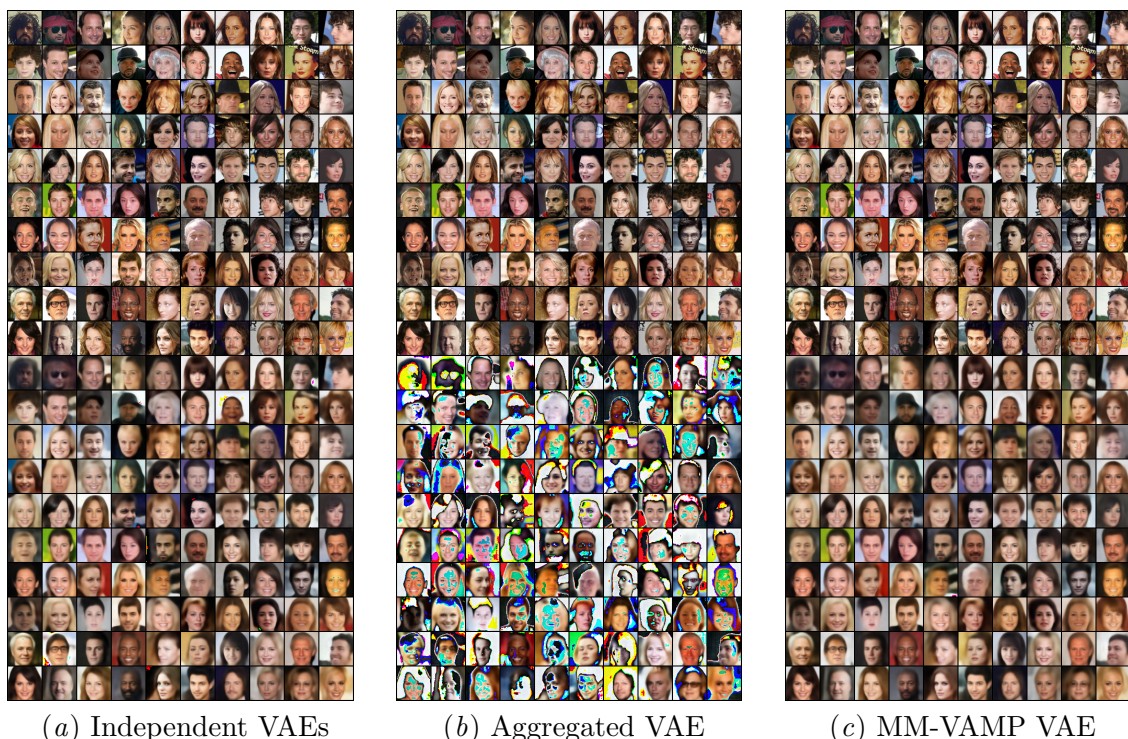

(a) Independent VAEs     (b) Aggregated VAE     (c) MM-VAMP VAE

Figure 10: Qualitative Results for the CelebA dataset on the image-to-image generation task. The first 10 rows of every subplot show the input image and the bottom 10 rows its conditional generation. Different to the training, we provide only the image to every model and based on the latent representation of that image, we generate a sample. We see that the aggregated VAE (fig. 10(b)) is not able to conditionally generate visually pleasing samples compared to the independent VAEs (fig. 10(a)) and the MM-VAMP VAE (fig. 10(c)).

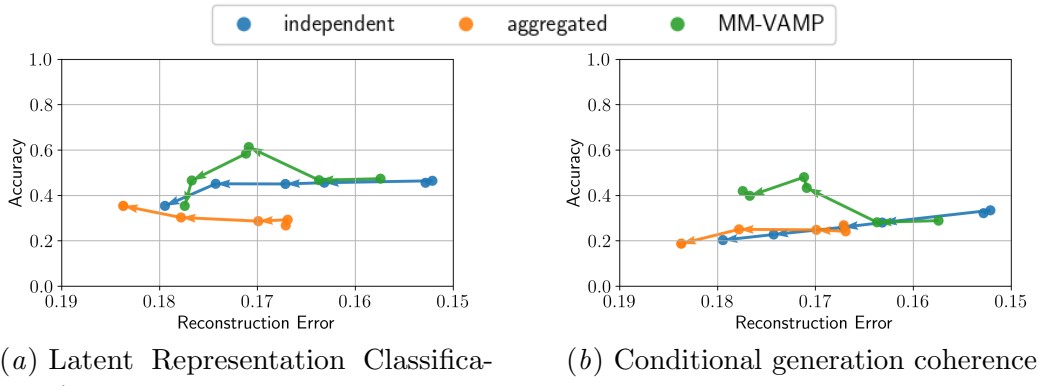

(a) Latent Representation Classification

(b) Conditional generation coherence

Figure 11: Results based on a memory experiment conducted on five rats, each regarded as a separate view. We report the performance of the latent representation classification and the conditional generation coherence against the reconstruction loss for different $\beta$ values for three different VAE methods. Every point in the figures represents a specific $\beta$ value, where $\beta = (10^{-5}, 10^{-4}, 10^{-3}, 2.5 \times 10^{-3}, 5 \times 10^{-3}, 10^{-2})$. An optimal model would be in the top right corner.

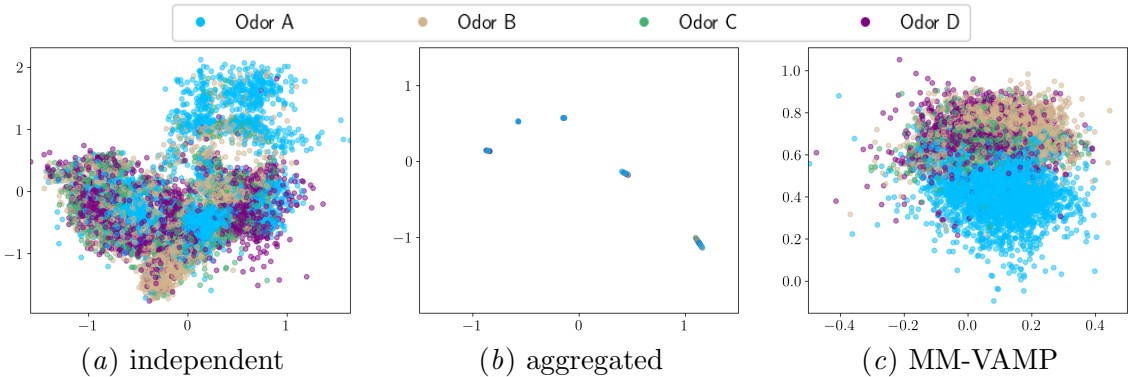

(a) independent

(b) aggregated

(c) MM-VAMP

Figure 12: Latent neural representation during a memory experiment. Each model's performance is evaluated based on its own optimal $\beta$ value (0.00001, 0.01, 0.001 for independent, aggregated, and MM-VAMP respectively) in terms of the self-conditioned latent representation classification accuracy according to Figure 11(a). Our model can distinguish the odor stimuli in the latent space with a clear separation of odors (4 different colors). Conversely, the two baseline models failed to combine multi-views as the odor separation only occurred within single views.

### C.5. Hippocampal Neural Activities

#### C.5.1. DATASET

Temporal organization is crucial to memory, affecting various perceptual, cognitive, and motor processes. While we have made progress in understanding the brain's processing of the spatial context of memories, our knowledge of their temporal structure is still very limited. To this end, neuroscientists have recorded neural activity in the hippocampus of rats performing a complex sequence memory task (Allen et al., 2016; Shahbaba et al., 2022). More specifically, this study investigates the temporal organization of memory and behavior by recording neural activity from the dorsal CA1 region of the hippocampus. Briefly, the task involves presenting rats with a repeated sequence of non-spatial events (four stimuli: odors A, B, C, D) at a single port. On average, about 40 of the recorded neurons (per rat) are active throughout the experiments. For those trials, the spiking activity is extracted during the time window of $(100, 400)$ milliseconds (ms) after the rat's nose enters the port. This window has been chosen to focus on the time period in which hippocampal activity primarily reflects the processing of the incoming odor (Shahbaba et al., 2022).

For such experiments, the systematic heterogeneity across individuals (e.g., biological and cognitive variability) can significantly undermine the power of existing methods in identifying the underlying neural dynamics. Since the same experimental setup was conducted across all rats, we consider the rats as different "modalities" and apply our proposed MM-VAMP VAE to capture the shared latent representation. While the existence (and importance) of subject-specific effects is well-known in neuroscience, such effects tend to be treated as unexplained variance because of the lack of the required analytical tools to extract and utilize this information properly. Our approach addresses this issue by providing a natural framework for sharing information across different subjects, which would otherwise be challenging.

The training data was collected from 250 ms length time frames after the port entry. Due to the behavior difference from each rat (some rats react faster to the odors while some others react slower), the training time frames of the five rats started from 250 ms, 250 ms, 500 ms, 500 ms, and 250 ms, respectively. During training, we treated each data point as independent and trained all the VAE models based on sliding windows (100 ms sub-window, 10 ms steps; 16 data points per window on each trial). The 100 ms sub-windows constituted the input data, with the dimension equal to the rat's number of neurons multiplied by 10, as the data was further binned into 10-ms increments.

#### C.5.2. IMPLEMENTATION & TRAINING

We use the same network architectures for all multimodal VAEs. Each of the autoencoders includes its unique encoder and decoder, both containing two hidden layers, without weight-sharing during training and evaluation. All modalities share the same architecture but the layers' dimensions are different, with 920, 790, 1040, 490, 460 dimensional input and hidden layers, respectively. The activation function was chosen to be LeakyReLU with a 0.01 negative slope. For all experiments on this dataset, we use an Adam optimizer with an initial learning rate of 0.001, a batch size of 128. We train all models for 1000 epochs.

### C.5.3. Results

Our proposed MM-VAMP model behaved the best at latent representation learning and conditional generation coherence among the three models. We trained the three VAE models on the neural data collected from the five rats. Figure 11($a$) shows the separation of the latent representation in terms of self-conditioned reconstruction (measured by the accuracy of a multinomial logistic regression classifier) against the reconstruction loss. Similar to the results of PolyMNIST, the proposed MM-VAMP VAE outperforms the two baseline models by a large margin by providing a clear separation of odors in the latent space while maintaining a low reconstruction loss. Figure 11($b$) compares the coherence of conditional generation accuracy against the reconstruction loss. As before, our proposed model outperforms the alternatives.

Another advantage of MM-VAMP is that it allows learning a shared latent representation across different modalities. We show the 2-dimensional self-conditioned latent representations through three VAE encoders in Figure 12 and Figure 13. In these two figures, each dot is the two-dimensional latent representation of a 100 ms sub-window of one odor trial for one rat. Figure 12 is colored by 4 odors, and Figure 13 is colored by 5 modalities (rats). Figure 12 shows the odor stimuli separation on the latent space and how good MM-VAMP VAE is in separating the odors. Figure 13 shows that the proposed MM-VAMP VAE can best utilize the shared information between the five rats by pulling the latent representations together. At the same time, two baseline models fail to extract the shared information between rats. Although showing separation in some views, the independent model did not provide a connection between views. The five tiny clusters in Figure 12 and Figure 13 show that, instead of showing a clear odor separation on the latent space, the aggregated model separated the data by rats. The results went against the intention to share information across views. In other words, the five rats' latent representations were far away from each other, so the aggregated VAE completely failed to connect the five views.

The qualitative results of the latent representation are shown in Figure 12 and Figure 13. In these two figures, each dot is the two-dimensional latent representation of a 100 ms sub-window of one odor trial for one rat. Figure 12 is colored by 4 odors, and Figure 13 is colored by 5 modalities (rats). Figure 12 shows the odor stimuli separation on the latent space and how good MM-VAMP is in separating the odors. Figure 13 shows that the proposed MM-VAMP VAE can best utilize the shared information between the five rats by pulling the latent representations together. At the same time, two baseline models fail to extract the shared information between rats. Although showing separation in some views, the independent model did not provide a connection between views. Because the five rats' latent representations were far away from each other, the aggregated VAE completely failed to connect the five views.

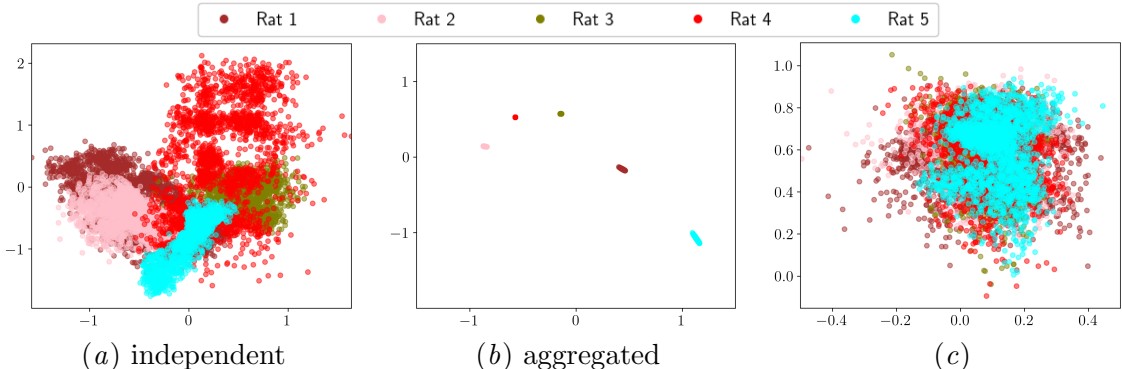

(a) independent      (b) aggregated      (c)

Figure 13: Latent Representation of Rats Brain by Each Rat. In our proposed MM-VAMP model, the five views shared latent representations as the latent representation of all five views (colors) gathered together, while the two baseline models failed to combine multi-views. As shown in the baseline models' latent representation, the five clusters (colors) kept away from each other and did not share information.

