# OpenReview forum: "Unity by Diversity: Improved Representation Learning in Multimodal VAEs"
_approximateinference.org/AABI/2024/Symposium — AABI 2024_

### Official Review · Reviewer_b3DG · 2024-04-24
**Interesting paper, more clarity can make it stronger**

**Rating:** 6
**Confidence:** 3

**Review:**

The paper proposes a new way to construct a multimodal VAE that does not rely on a hard constraint to aggregate the different modalities latent representations into a unique representation but instead, they use a soft constraint by defining a common prior for all the modalities inspired by the VAMP prior paper. The prior consists of a mixture of variational distributions obtained from the different modalities, meaning that the components of the mixture depend on the different inputs. They present results on three different multimodal datasets: PolyMNIST, bimodal Celeba, and a Hippocampal Neural Activities dataset. Results show that the proposed method learns a better latent representation and can generate more coherent conditional samples than two baselines, one that consists of independent VAEs and one that considers a simple way to aggregate the different latent representations from the different modalities into a single latent.

**Strength**
- The idea to favor an aggregated representation for different modalities by imposing a soft constraint using the prior is neat and clear. While the same idea is used differently in the context of prior and aggregated posterior mismatch, the application to multimodal VAE is new.
- There is an extensive evaluation of the method using three different datasets.
- The paper is well written.

**Thing that can improve the paper**
- The paper is well written, although I feel like the clarity can be improved and the overall message could benefit from that. Maybe the authors disagree with me, and it might be because I am not a complete expert in multimodal VAE. I think a central role in the evaluation is played by how one can sample missing modalities conditioned on the observed one. I think explaining how this is done with the three different models you are considering would be helpful. Related to this, in the bimodal Celeba experiment (Fig 10) you are presenting results on image-to-image, I agree that it's still conditional sampling, but I would highlight the fact that it is just reconstruction and not sampling a missing modality in that setting.
- I have some minor comments that are the results of trimming the paper down: after Eq 1 you are mentioning different lines but the equation consists of a unique line. The last paragraph of Section 1 is missing an 'Instead' at the beginning. Appendix B1 missing parenthesis, and appendix B3 has some additional parenthesis.
- You write that all the models get better reconstruction when $\beta = 0$, but isn't it the case most of the time in VAEs since  $\beta = 0$ means no KL, so in that case, the model is trained to just minimized the reconstruction error?
- As mentioned before, not a full expert on multimodal VAE, but you might consider different aggregation methods as baselines if exist, to make the paper even stronger.

---

### Official Review · Reviewer_phyM · 2024-04-25
**The paper proposes a novel architecture, the MM-VAMP VAE, for learning better representations in multimodal data, which is supported by solid theoretical evidence and validated through extensive experiments.**

**Rating:** 7
**Confidence:** 3

**Review:**

The main contribution of this paper is that it proposes a novel multimodal VAE called MM-VAMP VAE, in which a data-dependent prior distribution is proposed to enable a soft-sharing of information between different modalities. This approach overcomes the limitations of previous methods, which were sub-optimal and over-restrictive, such as those aggregating methods through a joint posterior approximation. The paper clearly illustrates the motivation for using a data-dependent prior and gives credit to the inspirational paper on the Vamp-prior. Furthermore, the paper supports the design of the architecture by conducting extensive experiments on three datasets. Due to page limits, I only reviewed the results of the PolyMNIST dataset in the main paper.  These findings are promising and suggest that the proposed model could significantly enhance multimodal data interpretation.

I have a couple of queries that I believe could further clarify the paper’s contributions and assumptions:

1. The authors claim in Section 1 (Introduction) and Section 2 (Multimodal VAEs - Encoder and Decoder) that the proposed method assumes that the likelihood and posterior entropy terms decouple across modalities and aims to balance the amount of shared and modality-specific information in the latent representation. I am wondering if you have supported this claim somewhere in the paper, either empirically or theoretically? How do you evaluate that your method can better balance these two kinds of information?
2. It is a bit vague to me when the authors talk about modality space and data space. It would be nice if the authors could clarify the difference between them using some specific problems.

---

### Official Review · Reviewer_QGXo · 2024-04-25

**Rating:** 7
**Confidence:** 4

**Review:**

In this paper, the authors propose a class of variational autoencoders called MM-VAMP VAE, which imposes a multimodal mixture-of-experts prior over based on the data. They showed that their models can learn more meaningful representations of miss data modalities.

In general this paper is well structured, and to my best knowledge, the math derivations are sound. Though I believe there is a line of work on add multimodal prior in VAEs, I still find this work highly motivating.

My main concern is about benchmark. I wonder if the authors can compare with a few more variants of VAEs such that we can see more comprehensive evaluations on empirical results.

---

### Official Review · Reviewer_4bhh · 2024-04-25
**Unity by Diversity...Review**

**Rating:** 7
**Confidence:** 4

**Review:**

Overview:

This paper proposes a 'mixture of experts' to develop a variant of auto-encoder that uses several parallel auto-encoders combined by a mixture distribution in the latent space.

Overall, the paper has some exciting ideas but needs some minor revisions to be publication ready.

Main Comments:

Some of the language is unclear. The terms "softly guiding", "sot constraints", "soft-alignment" and "soft sharing" and "soft-sharing" are all used interchangeably. Please unify these terms and clearly define what they mean.

The architecture proposed by this paper is very reminiscent of Chart-Autoencoders [1,2,3], but the authors do not reference recent work on geometric/topological auto-encoding. The authors reference a common problem in geometric auto-encoding in at the beginning of section 2 (multiple camera angles of a single object forming a low-dimensional manifold in the high-dimensional pixel space) but do not revisit the problem in the numerical examples.

[1] Schonsheck, Stefan, Jie Chen, and Rongjie Lai. "Chart auto-encoders for manifold structured data." arXiv preprint arXiv:1912.10094 (2019).
[2] Floryan, Daniel, and Michael D. Graham. "Data-driven discovery of intrinsic dynamics." Nature Machine Intelligence 4.12 (2022): 1113-1120.
[3] Loaiza-Ganem, Gabriel, et al. "Deep Generative Models through the Lens of the Manifold Hypothesis: A Survey and New Connections." arXiv preprint arXiv:2404.02954 (2024).

Minor Comments:

There are several typing setting errors involving incorrect use of quotation marks. Please do a global search and correction.

The term KL clearly refers to Kullback-Leibler, but it must be properly defined/introduced before it is used as an abbreviation.

In the paragraph after equation 1 there are references to 'first line' and 'second line' which do not make sense in the current typesetting.

Equation 3 (bottom half of page 4) does not have a numbered reference.

The term "R" is used in the narrative before being formally defined. Its definition is evident from context, but it should be included in eqn (1) before appearing in eqn (3). Similarly, the term Beta only appears in the narrative portion but is not included in any offset equations.
"Lemma 1" is mislabeled and should be named "Theorem 1."

---

### Official Review · Reviewer_3t8E · 2024-04-26

**Rating:** 6
**Confidence:** 3

**Review:**

Summary
This paper introduces the Multimodal Variational Mixture-of-Experts (MM-VAMP) VAE, an approach to learning from multimodal data. Unlike previous methods that enforced rigid joint representations across modalities, MM-VAMP uses a mixture-of-experts prior that softly guides each modality's latent representation towards a shared aggregate posterior. The method was evaluated using multiple datasets, including PolyMNIST and a bimodal CelebA dataset, and demonstrated improvements over baseline multimodal VAE models in terms of latent representation quality and generative quality .

Strengthens:
1. The MM-VAMP VAE introduces a soft constraint approach, allowing each modality to retain rich, individual information while still contributing to a coherent joint representation .
2. Across various datasets, MM-VAMP VAE outperforms traditional multimodal VAEs in learning meaningful latent representations and achieving higher generative coherence without compromising on reconstruction accuracy .

Disadvantages:
1. Complexity in Implementation and Tuning: It requires multiple hyperparameters and mixture components could make the model complex to implement and optimize. How do you solve the unbalanced capacity or the domination issue of each expert?
2. There is a paper demonstrating advanced Multi-modal VAE is not better than the union of uni-VAEs on high-dimensional data except CV tasks, e.g., gene expressions and molecular.   To support the claim, it's better to conduct experiment on diverse tasks.

---

### Official Review · Reviewer_AE6b · 2024-04-26
**Review for Unity by Diversity: Improved Representation Learning in Multimodal VAEs**

**Rating:** 7
**Confidence:** 4

**Review:**

The paper introduces an extension to variational autoencoders (VAEs) for multimodal data, which is critical for tasks like representation learning, conditional generation, and data imputation. Traditional multimodal VAEs often struggle with hard constraints that limit the flexibility of the model, primarily when sharing encoder or decoder components across different data modalities.

To address these challenges, the authors propose a novel method, the Multimodal Variational Mixture-of-Experts (MM-VAMP) VAE, which replaces the hard constraints of shared latent spaces with a soft constraint approach (eq 2 and Figure 1c). This approach utilizes a mixture-of-experts prior (based on the ideas of Tomczak and Welling (2017) ), enabling each modality to guide its latent representation towards a shared aggregate posterior softly. This technique allows each data modality to maintain more of its original feature information, enhancing the model's ability to manage and represent diverse data types effectively.

The paper provides a robust theoretical foundation for the proposed method and demonstrates its superiority through experimental evaluations on benchmark datasets. The results show significant improvements in learned latent representations (Figure 2).

Strengths of the work include:

- Innovative Methodology: The introduction of a mixture-of-experts prior for soft information sharing across modalities is an interesting innovation. It allows for flexibility and robustness in handling diverse and complex data structures.

- Practical  Relevance: The paper showcases improvements over traditional methods across several datasets. The use of real-world neuroscience data to validate the approach adds practical relevance and demonstrates the method's applicability in critical fields.

- Detailed Analysis: The authors provide a thorough theoretical exploration of their approach, including discussions on the optimality of the MM-VAMP prior and its advantages over hard constraint methods in terms of flexibility and information preservation.

Some weaknesses are:

- Generalization Concerns: While the model performs well on the datasets tested, the paper could explore more about its generalizability to other types of multimodal data not covered in the experiments (e.g. Mnist-SVHN,CUB (Shi et al., 2019) and MHD [1])

- Generative Capabilities. The analysis is satisfactory concerning the usefulness of the latent representations and the reconstruction errors of the VAEs. However, the validation is incomplete in terms of quantification of the generation quality: in particular, the authors could include standard metrics like the FID score to evaluate the goodness of the generated data (what is for example the FID of images in Figure 10?). I would also advise the authors to expand more on the sentence "We stress that by making the prior data dependent, our model no longer allows for unconditional generation; however, this property can be restored by incorporating hyperpriors"

- Clarity: is Theorem 1 just Lemma 1? Why do the authors mention the neuroscience experiment in the abstract and main paper if results are only reported in the appendix? In Figure 13c is probably missing a caption.

[1] Vasco et al. "Leveraging hierarchy in multimodal generative models for effective cross-modality inference". Neural Netw. 2022,

---

### Meta-Review · Area_Chair_zNub · 2024-05-24

**Recommendation:** Accept (Poster)
**Confidence:** 4

**Metareview:**

This paper proposes a novel multi-modal VAE formulation. All of the reviewers agreed that the paper should be accepted, with most finding it to be clear, original, and significant. No major issues with the soundness of the paper were brought up. Several minor clarity issues were pointed out, but these should be easily fixed for the camera-ready.

---

### Decision · Program_Chairs · 2024-05-27

Accept